# Spatial biases reduce the ability of earth system models to simulate soil heterotrophic respiration fluxes

Bertrand Guenet[1,*], Jérémie Orliac[1], Lauric Cécillon[1,2], Olivier Torres[3], Laura Sereni[4], Philip A. Martin[5], Pierre Barré[1], Laurent Bopp[3]

[1] Laboratoire de Géologie, Ecole normale supérieure, CNRS, IPSL, Université PSL, 24 Rue Lhomond, 75005 Paris, France

[2] Laboratoire ECODIV, Univ. Normandie, UNIROUEN, INRAE, Rouen, France.

[3] LMD-IPSL, Ecole Normale Supérieure, Université PSL, CNRS, Ecole Polytechnique, Sorbonne Université, Paris, France.

[4] INRAE, AgroParisTech, Université Paris-Saclay, UMR1402 ECOSYS, Ecotoxicology Team, 78026, Versailles, France.

[5] Basque Centre for Climate Change (BC3), Edificio sede no 1, planta 1, Parque científico UPV/EHU, Barrio Sarriena s/n, 48940, Leioa, Bizkaia, Spain.

*Correspondence to*: Bertrand Guenet (guenet@geologie.ens.fr)

**Abstract.** Heterotrophic respiration (Rh) is, at a global scale, one of the largest $CO_2$ fluxes between the earth's surface and atmosphere and may increase in the future. The previous generation of Earth System Models (ESMs) were able to reproduce global fluxes relatively well, but at that time no gridded products were available to perform an in-depth evaluation. The capacity of the new generation of ESMs used within the Coupled Model Intercomparison Project Phase 6 (CMIP6) to reproduce this flux has not been evaluated, meaning that the realism of resulting $CO_2$ flux estimates is unclear. In this study, we combine recently released observational data on Rh and ESM simulations to evaluate the ability of 13 ESMs from CMIP6 to reproduce Rh. Only four of the 13 tested ESMs were able to reproduce the total Rh flux but spatial analysis underlined important bias compensation for most of the ESMs which generally showed an overestimation in tropical regions and an underestimation in arid regions. To identify the main drivers of the bias, we performed an analysis of the residuals and found that mean annual precipitation was the most important driver explaining the difference between ESM simulations and observation-derived

product of Rh with higher bias between ESM simulations and Rh products where precipitation was high. Based on our results, next-generation ESMs should focus on improving the response of Rh to soil moisture.

## 1 Introduction

Soil organic carbon stocks represent around three times the amount of carbon in the atmosphere (Scharlemann et al., 2014). This soil carbon is used as a substrate by soil microorganisms to obtain their energy and feed their metabolism, which account for the majority of heterotrophic soil organism biomass. Annual fluxes that result from the respiration of these heterotrophic organisms (hereafter referred to as heterotrophic respiration) are estimated (Warner et al., 2019; Hashimoto et al., 2015; Konings et al., 2019; Ciais et al., 2021) to be five times higher than annual anthropogenic emissions (Friedlingstein et al., 2022) and roughly similar to annual terrestrial net primary production (Zhao et al., 2005). Thus, due to the size of fluxes relating to heterotrophic respiration, even minor changes in soil organic carbon dynamics can lead to significant impacts on carbon feedbacks and, ultimately, on climatic changes. As a result, modification of soil organic carbon stocks due to human activities is considered to be an  important driver of future climate trajectories (Chabbi et al., 2017).

Although heterotrophic respiration fluxes are important, the way this flux is represented in Earth System Models (ESMs), which aim to simulate the most important drivers of the earth's climate system, is currently challenged. This is because important drivers are missing and new approaches have been proposed (Huang et al., 2021; Wieder et al., 2015) they lack sufficient evaluation on long term time series (Le Noë et al., 2023). Thus, the identifying the accuracy of predictions of heterotrophic respiration fluxes by ESMs is a key topic in order to help constrain carbon-climate feedbacks in ESMs. The ability of ESMs to reproduce this flux was previously studied by Shao et al., (2013) but this was done using the previous generation of ESMs using the simulations done during the Coupled Model Intercomparison Project Phase 5 (CMIP5). Moreover, at that time no gridded products were available. Since then, the models have been greatly improved and assessing how accurately current ESMs reproduce the fluxes associated with heterotrophic respiration is therefore of major importance. Until now, it was not possible to undertake a robust spatial assessment because of the lack of observation-derived gridded products of Rh. In recent years, new gridded products derived either from (i) upscaling of local observation or (ii) calculations using atmospheric inversions and satellite observations. These products provide the opportunity to evaluate the simulations of ESMs used within the Coupled Model Intercomparison Project Phase 6 (CMIP6) against observation-derived products for heterotrophic respiration. CMIP is a key initiative which aims to compare current ESMs and is a central element of national and international assessments of climate change (Masson-Delmotte et al., 2021).

In this study we have two major aims:

1. Compare predictions of the total flux of heterotrophic respiration from 13 earth system models with three recent gridded products of heterotrophic respiration derived from observations and identify the spatial biases of heterotrophic respiration in the models.

2. Identify the major drivers of the heterotrophic respiration bias in earth system models to propose new developments for the next generation of earth system models using a model residue approach to disentangle the main effect.

## 2 Materials and Methods

### 2.1 Earth System Models simulations.

In this study, we used the model outputs from the 6[th] Coupled-Model Intercomparison Project (CMIP6) (Eyring et al., 2016) which coordinates global climate model simulations of the past, current, and future climate. CMIP6 proposes historical simulation spanning from 1850 to 2014. Historical simulations are driven from an initial point chosen in control integration (*piControl*). We chose to use the latest CMIP6 results for basic initial state (r1i1p1f1). We chose outputs from thirteen ESMs that provide heterotrophic respiration fluxes (BCC-CSM2-MR, BCC-ESM1, CanESM5, CESM2, CNRM-ESM2-1, E3SM-1-1-ECA, IPSL-CM6A-LR, MIROC-ES2L, MPI-ESM1-2-LR, NorCPM1, NorESM2-LM, SAM0-UNICON and UKESM1-0-LL). The variable used is "rh" corresponding to the total heterotrophic respiration on land. We computed annual average over the 1990-2010 period which corresponds to the period in which most of the observations in the global Soil Respiration Database (Bond-Lamberty and Thomson, 2010) v3.0 were made. Two observation products we used were obtained using those data.

### 2.2 Observation-derived products.

In this study we used three observation derived products (Warner et al., 2019; Konings et al., 2019; Hashimoto et al., 2015). In Warner et al. (2019), the authors predicted annual soil respiration and associated uncertainty across terrestrial areas at a resolution of 1 km using a quantile regression forest algorithm trained with observations from the global Soil Respiration Database (Bond-Lamberty and Thomson, 2010) v3.0 (commit number 651770 in GitHub, https://github.com/bpbond/srdb) spanning from 1961 to 2011 but mostly after 1990. Then they deduced Rh from the soil respiration using two different methods (Bond-Lamberty et al., 2004; Subke et al., 2006). They therefore proposed two Rh maps derived from a unique mean map of Rs from quantile regression forest model. Here, we decided to use the mean of two approaches as a reference for Warner et al. (2019) Rh results. The second product we used Hashimoto et al. (2015) is also based on the Soil Respiration Database (Bond-Lamberty and Thomson, 2010) v3.0 but in this case they derived the Rh flux using a climate-driven model of soil respiration derived from the Raich's model (Raich et al., 2002). They provided a 0.5∘ resolution product at a monthly step time between 1965 and 2012. In our case, we used the yearly average over the period. The third product used, Konings et al. (2019), estimated Rh as a residual remote-sensing data exploiting recent advances in carbon-flux estimations. In contrast with the two other products which are bottom-up, the Konings et al., (2019) product proposes a top-down approach combining net ecosystem productivity estimates from atmospheric inversions with an optimally scaled gross primary productivity dataset derived from satellite observations. Rh is then derived using the CARbon DAta MOdel fraMework, (CARDAMOM). Their result is a monthly evaluation of Rh, between January 2010 and December 2012, at a resolution 4°×5°.

## 2.3 Data treatment and regridding.

Since the ESMs outputs and products were not at the same resolution, we chose a reference for map-grid resolution. The coarsest resolution was from Konings et al.'s (2019) product with a 4°×5° resolution grid. Reduced the resolution of every Rh map to match this would cause a substantial loss of information. Thus, we increased the resolution of those datasets and decreased the very fine scale maps to an arbitrary reference corresponding to the CNRM-ESM2-1 model which runs at 0.7° resolution. We chose to set the reference at the maximum resolution available among CMIP6's ESMs predicting Rh. We used the common regridding routine Climate Data Operators (CDO) remapdis (nco module) that performs regridding by distance-weighted average remapping and conserve latitudinal and longitudinal means. The CDO software is a collection of multiple operators for standard processing of climate and forecast model data. The operators include simple functions (statistical and arithmetic) to be used for data selection, subsampling, and spatial interpolation. To avoid coastal pixels encroaching into oceans, we weighted each pixel by the proportion of its are covered by land. The sum of Rh over the lands was compared before and after regridding to ensure that it was conservative. When comparing the original and the regridded version of the Konings et al. (2019)'s product we observed very similar pattern (Fig. 1).

## 2.4 Comparison between models' outputs and heterotrophic respiration products.

To estimate the ability of the CMIP6's model to reproduce soil heterotrophic respiration, we first compared the global flux summed over all the grid cells and averaged over 1990-2010 period in Pg C yr$^{-1}$. We also compared the Rh maps after regridding, averaged over the 1990-2010 period. We also performed latitudinal and longitudinal means calculus including oceanic zero-values. Secondly, we wanted to assess spatial bias distribution. Therefore, we i) compared CMIP6 model average with observation products and ii) compared each CMIP6 models with observation products. Thus, we first represented the model average (over the period 1990-2010) and all the observation derived products on a same figure with their associated latitudinal and longitudinal means. We also calculated the 25th and 75th quantiles of latitudinal and longitudinal CMIP6 model means. Then, we computed the difference for each individual CMIP6 model with the median of the three observation products. To compare the ESMs with the observation products, we calculated the root mean square error (RMSE) and the $R^2$ using the median of the observation products. Finally, we also calculated the median absolute deviation (MAD) for each grid cells and we calculated the number of pixels for each model that fit within the median ± MAD.

## 2.5 ESM's model residual analysis.

We defined the ESM's model residuals as the median of the difference between each individual CMIP6's model output and the observation-based products median calculated for each grid cell. The ESM's model residuals were calculated in three steps:

1. We first calculated the median for each cell (i) that we called $Rh\_obs_i$ using the three observation-derived products using eq. 1 with *Rh_Hashimoto et al. (2015)$_i$, Rh_Warner et al. (2019)$_i$* and *Rh_Konings et al. (2019)$_i$* being the heterotrophic respiration given for the grid cell I by Hashimoto et al., (2015), Warner et al. (2019) and Konings et al. (2019, respectively. We consider this median as our best-estimate.

$$Rh\_obs_i = Med(Rh\_Hashimoto\ et\ al.\ (2015)_i, Rh\_Warner\ et\ al.\ (2019)_i, Rh\_Konings\ et\ al. (2019)_i) \quad (1)$$

2. Next, we calculated the residual between each CMIP6's model output and our best-estimate that we called $Res\_X_i$ (X being the model's name) for each grid cell i using eq. 2.

$$Res\_X_i = Rh\_X_i - Rh\_obs_i \quad\quad\quad (2)$$

3. Finally, we calculated the ESM's model residuals ($Res_i$) as the median of this model specific residuals ($Res\_X_i$) using eq. 3.

$$Rh\_X_i - Rh\_obs_i \quad\quad\quad (3)$$

Using the ESM's model residuals, we performed a statistical analysis to identify the main drivers of disagreement between predictions and observations. Following this we undertook a two-step methodology. First, we compared several linear generalized least square models with different spatial structures (gaussian, exponential, spherical, linear or rational (gls package, (Venables and Ripley, 2002))) and without spatial structures to estimate the effect of spatial correlation. Based on Akaike information criterion (AIC) values we selected the rational quadratic spatial correlation structure that had the smallest AIC values for the second step of the analysis. Then, we used generalized additive mixed model with ESM's model residuals as variable to explain and mean annual temperature (MAT), mean annual precipitation (MAP), observation derived SOC, ESM's model residuals on NPP and lithology as predictors variables. MAT and MAP are derived from the Global Soil Wetness Project Phase 3 (GSWP3) reanalysis (http://hydro.iis.u-tokyo.ac.jp/GSWP3/ last access: April 5 2022). SOC was taken from the Soilgrid250m product (Hengl et al., 2017). ESM's model residuals on NPP are calculated as the median of the difference between ESM's NPP and NPP from the global inventory monitoring and modelling studies group (GIMMS). Lithology maps from the global lithological map (GLiM) (Hartmann and Moosdorf, 2012) was used but since lithology was not significant ($p > 0.05$) and the model has a lower AIC without this variable, it was not included in the final generalized additive mixed model presented here. All statistical analyses were carried out using R v3.5 (R Core Team, 2018).

## 3 Results

### 3.1 Global heterotrophic respiration flux and spatial biases

Global heterotrophic respiration flux simulated by the 13 ESMs ranges from 29 to 78 Pg C yr$^{-1}$ (Fig. 2), whereas the equivalent estimates for observationally derived products estimate range from 43 to 51 Pg C yr$^{-1}$. The multi-model mean of the ESMs

(49 Pg C yr$^{-1}$) falls within the range of the observation-derived products. However, only four out of 13 ESMs (BCC-CSM2-MR, CNRM-ESM2-1, IPSL-CM6A-LR, and SAM0-UNICOM) simulate an overall heterotrophic respiration flux that is within the range of the observation-derived products (Fig 2). When comparing the model observation products' with the median ± MAD from the observation products (46 ± 7 Pg C yr$^{-1}$), seven out of the 13 ESMs predicted an heterotrophic respiration within this range (Fig. 2). The $R^2$ between the model outputs and the median of the observation products range between 0.57 for E3SM-1-1-ECA and 0.82 for MIROC-ES2L (Table 1). When using RMSE to compare the model outputs and the median of the observation products was 170.9 gC m$^{-2}$ yr$^{-1}$ for IPSL-CM6A-LR and 345.1 gC m$^{-2}$ yr$^{-1}$ for CanESM5 (Table 1). Finally, we also estimated the number of pixels that fell within the median ± MAD and using this metric BCC-ESM1 was the best performing model followed by BCC-CSM2-MR and CNRM-ESM2-1.

Despite similar global-scale values, regional-scale differences between the observation-derived products are much larger (Fig. 3). The Konings et al. (2019) product estimates large heterotrophic fluxes in the tropics and lower fluxes in other regions such as the west coast of Northern America or central Asia, as compared to the Warner et al. (2019) and the Hashimoto et al. (2015) products that share similar spatial patterns (Fig. 4). The mean of the 13 ESMs simulations also gives a much larger heterotrophic respiration fluxes over the tropics in particular over South-East Asia compared to any of the three observation-derived products. In general, the heterotrophic respiration fluxes from the 13 ESMs mean is closer to Konings et al. (2019) product over the tropics but closer to the Warner et al. (2019) and the Hashimoto et al., (2015) products over temperate regions. For boreal regions, the three observations-derived products and the 13 ESMs means are very close.

To generate our best-estimate of heterotrophic respiration fluxes from the three observation-derived products we calculated the median for each cell. Thus, we obtained the spatially distributed best-estimate. At each grid cell, we then compared each ESM with the observation-derived products median (Fig. 5). This evaluation indicates that, compared to observation-based products, ESMs (apart from the ESM NorCPM1) tend to overestimate heterotrophic respiration flux in tropical regions (approx. 1,000 gC m$^{-2}$ yr$^{-1}$ for MPI-ESM1-2-LR over the Amazon or 1,500 gC m$^{-2}$ yr$^{-1}$ for UKESM1-0-LL over South-East Asia, for instance). Models perform relatively well in temperate regions with bias close to 0 gC m-2 yr-1 for BCC-ESM-1 over North America and Europe. Important discrepancies were observed for boreal regions with some models underestimating the heterotrophic respiration fluxes to a large degree (e.g. NorCPM1 or SAM0-UNICON) and one overestimating the fluxes (MPI-ESM1-2-LR). The BCC models (BCC-CSM2-MR and BCC-ESM1) performed quite well over this region. Importantly, the four models that predict a global heterotrophic respiration flux within the range given by the observation-derived products (BCC-CSM2-MR, CNRM-ESM2-1, IPSL-CM6A-LR and SAM0-UNICOM), do not perform well at finer scales - with over-estimation of the flux in some regions and under estimation in others. Therefore, this good global-scale performance masks spatial bias compensation.

**3.2 Identification of the major drivers of the heterotrophic respiration bias in earth system models.**

185 In order to improve predictions of heterotrophic respiration fluxes in future ESMs we need to understand the spatial biases we observed and determine their causes. To explore these biases, we performed a statistical analysis based on a generalized additive mixed model of the ESMs residuals defined as the median of the difference between each CMIP6 model's output and the median of the observation-based products calculated in each grid cell. ESMs share a very common approach based on first order kinetics with soil organic decomposition driven by soil moisture and temperature (Varney et al., 2022; Todd-Brown et

190 al., 2014). This approach is derived from the very first attempts to describe soil organic decomposition with mathematical equations (Henin and Dupuis, 1945) and is still the most used to describe this process (Manzoni and Porporato, 2009; Wutzler et al., 2008). Since SOM decomposition schemes in ESMs are very similar, comparing each model individually can be redundant and not very informative and less generalizable. To allow broader conclusions and suggestions to improve ESMs performances, we decided to perform the residual analysis on the ESMs median rather on each individual model.

195 The main drivers of heterotrophic respiration are soil carbon availability, soil moisture and temperature, carbon inputs and mineralogy (Doetterl et al., 2015). To explain our model residues we used soil organic carbon, net primary production residuals calculated using similar methods to heterotrophic respiration flux residuals, mean annual precipitation, mean annual temperature and lithology. Our method identified the main drivers of ESMs residuals as soil organic carbon, net primary production residuals, mean annual precipitation, and mean annual temperature (Fig. 6). The effect of lithology was not

200 statistically significant ($p > 0.05$) and the model has a lower AIC without this variable and so we did not include lithology in the final model presented here. The residuals due to soil organic carbon stock are close to zero for soil with a low carbon stock but heterotrophic respiration is under estimated by ESMs for soils rich in organic carbon ($> 3,000$ g C m$^{-2}$) (Fig. 6a). The model residuals for heterotrophic respiration flux are partially explained by the model residuals on net primary production with a slight increase from model underestimation to model overestimation when model residual on net primary production

205 increase from -1,000 to 400 g C m$^{-2}$ yr$^{-1}$. We noted that when net primary production fits well with satellite products (i.e. model residuals close to 0 g C m$^{-2}$ yr$^{-1}$), the ESM residuals on the heterotrophic respiration flux are also close to 0 g C m$^{-2}$ yr$^{-1}$. For a few grid cells where ESMs largely overestimate net primary production (i.e. model residuals higher than 400 g C m$^{-2}$ yr$^{-1}$), the ESMs residuals on heterotrophic respiration flux tend to be negative suggesting that ESM underestimate heterotrophic respiration flux. The clearest tendency we obtained was with mean annual precipitation, the more it increases the more the

210 models overestimate the heterotrophic respiration flux (Fig. 6c). The median ESMs residual was also partially controlled by mean annual temperature (Fig. 6d) with a relatively low overestimation by the models for cold temperatures such as those recorded in polar climate zones and in some continental climate zones (e.g. subarctic climate), a relatively good fit for temperature between 0 and 20°C corresponding to temperate and some continental climate zones (e.g. Hot summer continental climates) and then a sudden underestimation for warm temperatures above 20°C corresponding to tropical and dry climate

215 zones. This sudden underestimation might be explained by an arbitrary maximum respiration level observed in this dataset and identified as the result of the temperature-dependence of soil respiration used by Hashimoto et al., (2015) (Varney et al., 2020).

Such bias may therefore be a consequence of the observation-based products used here rather than a real bias in ESMs. Similar results were obtained when performing the same analysis with means rather than medians (Fig. 7).

## 4 Discussion

In this study we evaluated, for the first time, the ability of the ESMs to reproduce heterotrophic respiration fluxes. Indeed, previous dataset were not gridded and so far spatial patterns of heterotrophic respiration in ESMs could only be constrained indirectly by constraining other C fluxes including heterotrophic respiration such as net ecosystem exchange fluxes or through ecosystem respiration in which heterotrophic respiration is just one component the other being the autotrophic respiration (Stoy et al., 2013). We showed that only four of 13 of the CMIP models produce global-scale estimates that are

consistent with observation-derived products. However, we also showed that this consistency was due to spatial bias compensations driven by different environmental variables. Heterotrophic respiration represents a carbon flux that is roughly five times that of anthropogenic emissions (Friedlingstein et al., 2022) and, as such, it is vital that work is done to improve the ability of ESMs to reproduce this flux. Nevertheless, we also observed large discrepancies between observation-based products showing that our ability to provide heterotrophic flux based on observations is not optimal. To better constrain ESMs

projections, some efforts are needed to reduce residuals between observation-based products.

However, working only on heterotrophic respiration may not be sufficient to improve the entire soil organic carbon module of the ESMs (Table 2). ESM capacities to reproduce observed soil organic carbon stocks also need to be improved (Ito et al., 2020; Varney et al., 2022). To improve both soil organic carbon stocks and heterotrophic respiration fluxes soil organic carbon decomposition rates must be better constrained. The ESM residual analysis we performed here suggests some new research

avenues relating to the response of the major drivers. First, most of the boundary conditions of the soil organic carbon modules of an ESM are calculated by the ESM itself. Thus, if soil moisture, soil temperature or litter production are incorrect, the soil organic carbon dynamic cannot be correct. We observed that when the residual of NPP was close to zero the residual on heterotrophic respiration is also close to zero. Thus, improving the plant functioning scheme may ultimately improve the capacities of the ESMs to reproduce the heterotrophic respiration flux. Our study also showed that mean annual temperature

is an important driver of the ESM residuals in particular for hot regions with large underestimations of the flux. It probably corresponds to very arid regions since for most of the ESMs, heterotrophic respiration fluxes from regions like Australia, the Middle East or Northern Africa tend to be underestimated. Nevertheless, the underestimation observed in these regions may be also due to reduced C inputs and low SOC stocks reducing the heterotrophic respiration fluxes.

The response of soil organic decomposition by microorganisms is likely to be temperature dependent, with lower rates of

decomposition seen in cold regions and higher rates in hot regions (Wang et al., 2010; Zhou et al., 2009). In contrast, the response of soil organic decomposition to temperature in ESMs is generally controlled by Q10 equations (Davidson and Janssens, 2006) with fixed parameters not dynamic and not spatially distributed (Ito et al., 2020). Previous studies suggested

that a spatially distributed Q10 constrained by observations are an important step to improve ESMs (Koven et al., 2017; Varney et al., 2020). Our results support this and hint that having more flexible Q10 parameters may help to improve ESMs capacities to reproduce observation-derived products of heterotrophic respiration fluxes. Moreover, land surface scheme of ESMs are known to be very sensitive to Q10 values (Jones et al., 2003; Todd-Brown et al., 2018).

Finally, we observed a relatively linear, positive relationship between mean annual precipitation and the ESMs' residuals (Fig. 6c). This response is probably driven by soil moisture because it is a key driver of microbial activity and therefore of heterotrophic respiration fluxes (Moyano et al., 2012). ESMs use three main groups of soil moisture response function (Falloon et al., 2011): i) some models do not represent soil moisture effect, ii) some models increase soil organic decomposition when soil moisture increases assuming less water limitation for microbial activity and iii) some models assume a humped relationship between soil moisture and soil organic decomposition, with high decomposition at intermediate soil moisture and low decomposition in very wet soils where microbial activity is reduced because of limitation by oxygen availability and in dry soils where microbial activity is reduced because of limitation by water. As with Q10, the land surface models are highly sensitive to which soil moisture response function chosen and most of the ESMs use option ii) (Varney et al., 2022). Soil incubations have repeatedly shown that the response of heterotrophic respiration fluxes to soil moisture is approximated by a bell-shaped function with parameters depending on soil organic carbon, soil clay content, and soil bulk density (Moyano et al., 2012). Thus, for wet soils, heterotrophic respiration fluxes are probably reduced because of oxygen limitation. Implementing this bell-shaped function approach is necessary to accurately represent the soil organic carbon stock of peatland in some land surface schemes used by ESMs (Qiu et al., 2019). The approach proposed by Moyano et al., (2012) seems well adapted to constrain ESMs since the author proposed several versions of the bell-shaped function and defined a function using drivers that are included in ESMs (the model 2 in Moyano et al., (2012)). The model including bulk density might perform better but bulk density is not calculated by ESMs and consequently such approach is hardly implementable in ESMs. Other approaches have been proposed in the literature (Davidson et al., 2014; Sierra et al., 2014) but the solutions proposed are mostly based on $O_2$ diffusion which is more mechanistic but more difficult to implement in an ESM compared to a more empirical solution as proposed by Moyano et al. (2012). Gas diffusion implementation at the spatial resolution of ESMs is quite challenging because it depends on drivers highly variables at small scales. Not considering the possible oxygen limitation effect on wet soils may explain why ESMs tends to overestimate the heterotrophic respiration flux when mean annual precipitation is high. Changing soil moisture function to better represent this effect should be relatively easy and may substantially improve the capacities of ESMs to reproduce the heterotrophic respiration flux.

Another important parameter controlling heterotrophic respiration flux is carbon use efficiency defined as the ratio between the carbon remaining in a system and the carbon entering that system (Manzoni et al., 2018). In our context this is the ratio between the carbon mineralized through microbial heterotrophic respiration and the carbon incorporated into the microbial biomass. The heterotrophic respiration flux therefore results from two processes in ESMs, the soil organic carbon decomposition and its allocation to other soil carbon pools or to heterotrophic respiration. Carbon use efficiency is highly

variable and depends on several biotic and abiotic factors (Sinsabaugh et al., 2013; Manzoni, 2017; Manzoni et al., 2012). In ESMs, carbon use efficiency is neither dynamic nor spatially distributed, and thus having flexible carbon use efficiency control may help to reproduce observations (Zhang et al., 2018). A simple approach that may aid a better representation of heterotrophic respiration fluxes is optimizing the carbon use efficiency parameters of the ESMs using a Bayesian approach as

is done for other land fluxes (Kuppel et al., 2012). This would result in a spatially distributed set of parameters for carbon use efficiency but this approach would not be dynamic. Another option that might benefit the current large carbon use efficiency measures existing in the literature (Manzoni et al., 2012) to define statistical functions predicting carbon use efficiency based on explanatory variables that could themselves be dynamic (soil temperature, pH, soil C:N ratio, etc.). Thus, carbon use efficiency might be spatialized and dynamic.

A better representation of the heterotrophic respiration flux is also important for other biogeochemical variables in particular in ESMs with explicit nitrogen cycle representation in their land surface scheme. Indeed, heterotrophic respiration fluxes are indicators of soil organic carbon decomposition but when nitrogen is explicitly represented it also becomes an indicator of soil N mineralization (Vuichard et al., 2018). In the field, the soil organic matter is composed by complex molecules made of carbon and nitrogen among others (Cleveland and Liptzin, 2007). Microorganisms decompose soil organic matter releasing

$CO_2$ to the atmosphere and mineral nitrogen to the soil solution. Microbial activity is therefore a major driver of mineral nitrogen availability and partially control nitrogen limitation on primary production and therefore on land carbon sink (Bragazza et al., 2013). Since more and more ESMs explicitly represent the nitrogen cycle in their land surface models (Varney et al., 2022; Davies-Barnard et al., 2020) resulting in well constrained heterotrophic respiration fluxes that may help to constrain the nitrogen mineralization flux as they both come from the soil organic matter decomposition by extracellular

enzymes. A better representation of the mineral N release flux would probably, in turn, improve the simulation of NPP.

**5 Conclusion**

Our study showed that despite previous ESMs evaluation of heterotrophic respiration (Shao et al., 2013), few current ESMs represented total heterotrophic respiration flux well and all failed at representing its spatial distribution. Since heterotrophic fluxes are large and are a major determinant of whether land surfaces represent a carbon sink or source it is of major importance

to better constrain these fluxes and how they will be impacted by climate and land use changes. We showed that current ESMs failed to reproduce heterotrophic respiration fluxes where precipitation is important probably because heterotrophic respiration responses to soil moisture are poor representations of reality. Nevertheless, it is important to note that soil moisture is not only driven by precipitation. Other water fluxes like runoff, drainage and evapotranspiration affect the water balance in soils. In this study we did not directly consider soil moisture because it was not available for all the ESMs. Another limitation of our

study is that we did not account for other important drivers of heterotrophic respiration in our model residual analysis like pH, microbial biomass, nitrogen availability, etc. We decided to focus on explanatory variables calculated by all the models because

we aimed to identify biases due to feedbacks between ESMs' variables rather than identifying missing mechanisms. We propose several options to improve the ESM without extensive modifications of the current schemes. We believe that our proposals would be relatively easy to implement in the next generation of ESMs resulting in possible improvements. Finally, we observed important discrepancies between the observation products we used. We decided to use the median as the best estimate here but to better constraint ESM in the future, observation-based products need to be more accurate. Increasing the number of observations used to create such gridded products is an obvious recommendation but this can be complicated. New satellite products like GOSAT or OCO-2 are measuring net ecosystem exchange (Palmer et al., 2019). These satellite observations represent good candidates to better constrain the observation products in the future.

**Data availability**

All data are available in the main text.

**Author contributions**

BG, LC, PB and LB designed the study, BG, JO, OT and LS performed the analysis. All the authors participated to the results interpretation and to the writing.

**Competing interests**

All other authors declare they have no competing interests.

**Acknowledgments**

The IPSL-CM6 experiments were performed using the HPC resources of TGCC under the allocations 2019-A0060107732, 2020-A0080107732 and 2021-A0100107732 (project gencmip6) provided by GENCI (Grand Equipement National de Calcul Intensif). The IPSL-CM6 team of the IPSL Climate Modelling Centre (https://cmc.ipsl.fr) is acknowledged for having developed, tested, evaluated, tuned the IPSL climate model, as well as performed and published the CMIP6 experiments. This study benefited from the ESPRI computing and data centre (https://mesocentre.ipsl.fr) which is supported by CNRS, Sorbonne Université, Ecole Polytechnique and CNES as well as through national and international grants. BG acknowledges the Climat AmSud program grant REPRISE 21-CLIMAT-13 and the Agence Nationale de la Recherche EASIER-F (ANR-21-ERCC-0008) for funding.

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

**Figures**

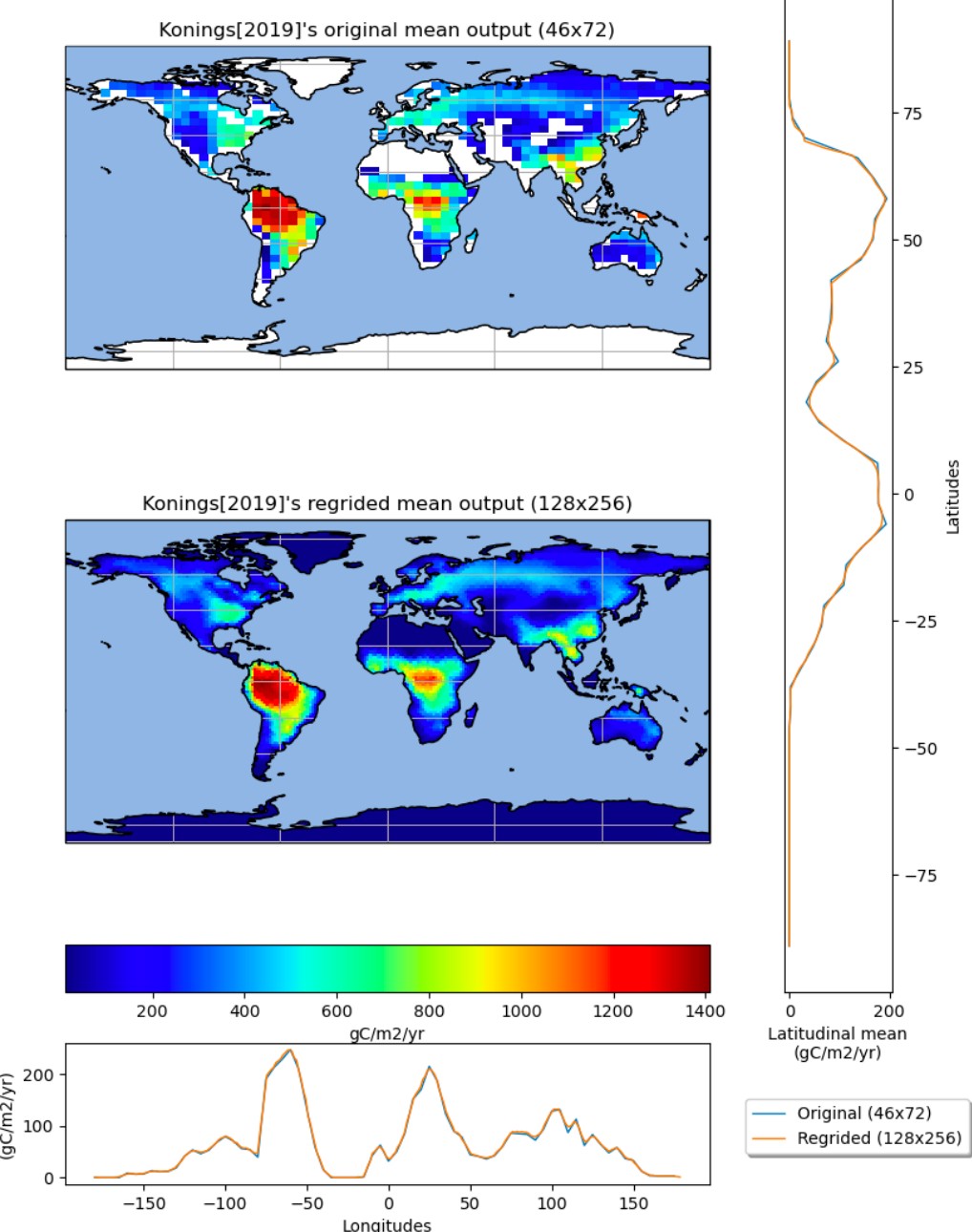

**Fig. 1. Mean Rh spatial distribution over 2010-2012 from the Konings et al., (2019) product –original (46x72, top panel) vs regrided (128x256, bottom panel).**

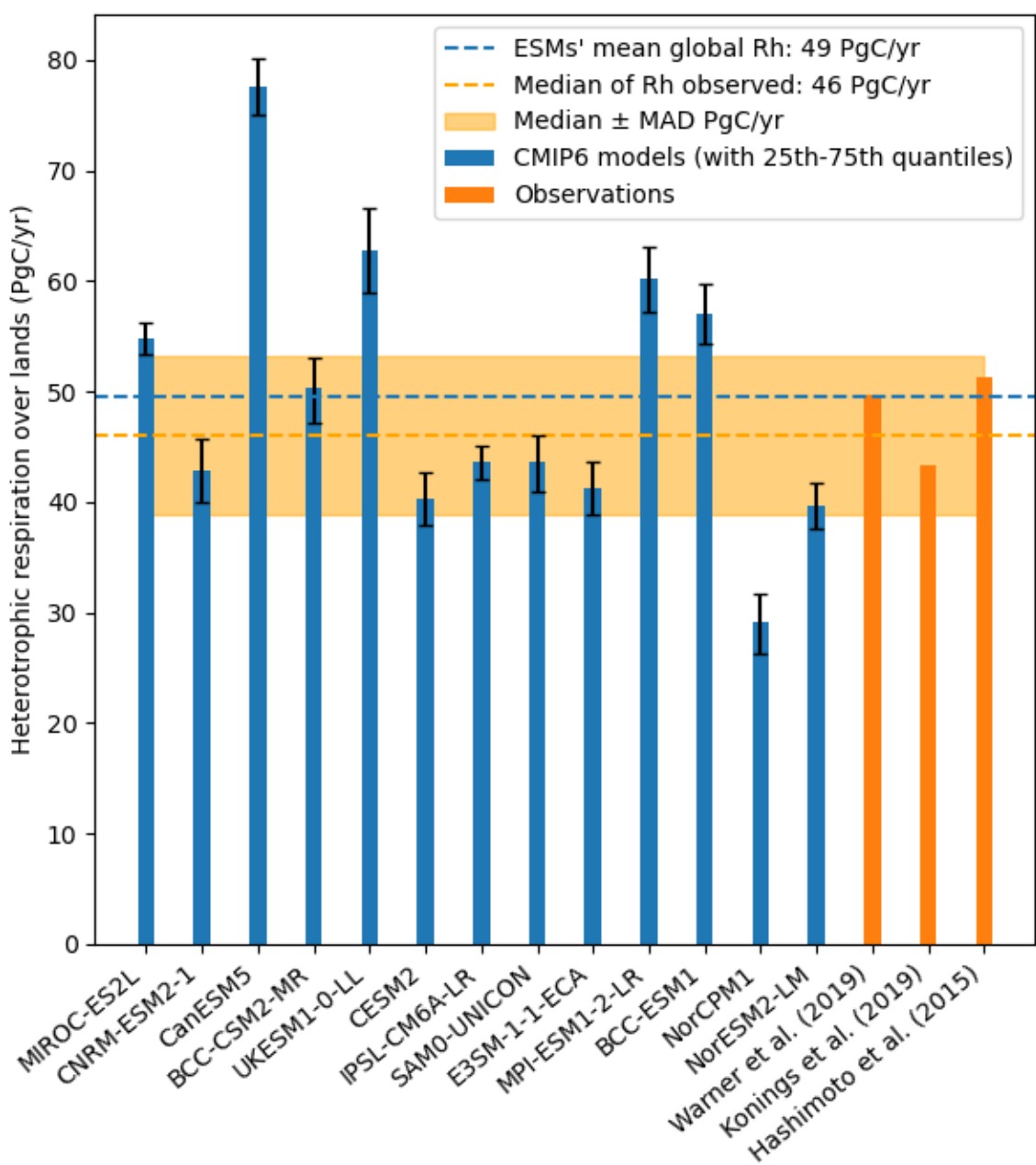

**Fig. 2. Global estimations of soil heterotrophic respiration mean over 1990-2010 period.**

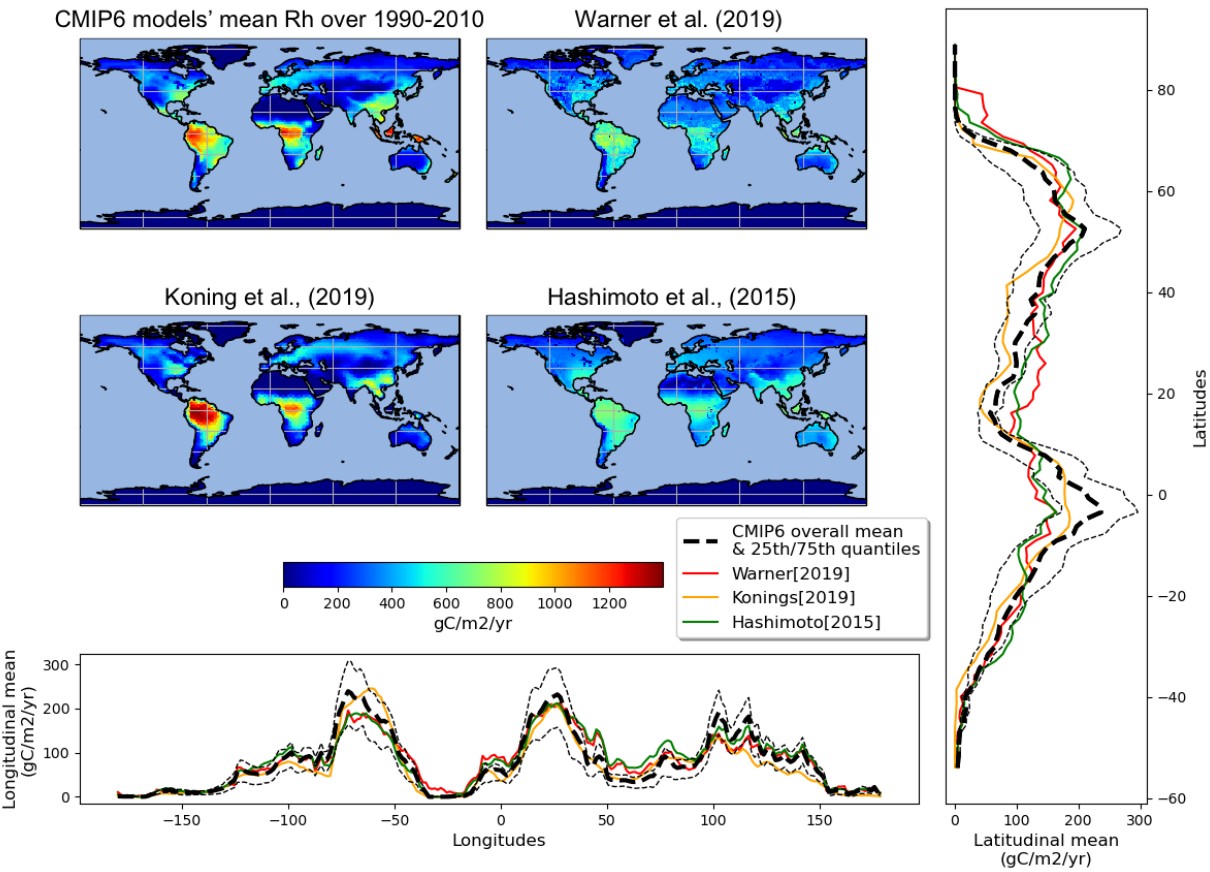

**Fig. 3. Comparison of mean soil heterotrophic respiration spatial distribution among mean CMIP6 outputs and observation data.**

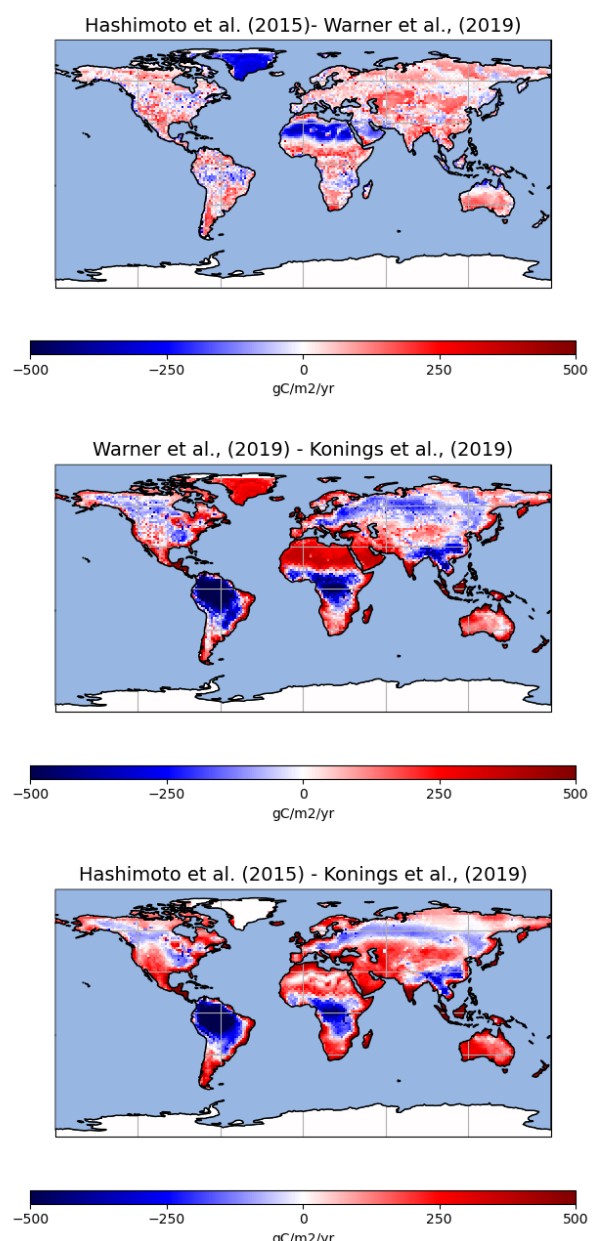

**Fig. 4. Maps of difference between the observation based product used in this study (Hashimoto et al., (2015) – Warner et al., (2019) in the top panel, Warner et al., (2019) – Konings et al. (2019) in the middle panel and Hashimoto et al., (2015) – Konings et al. (2019) in the bottom panel).**

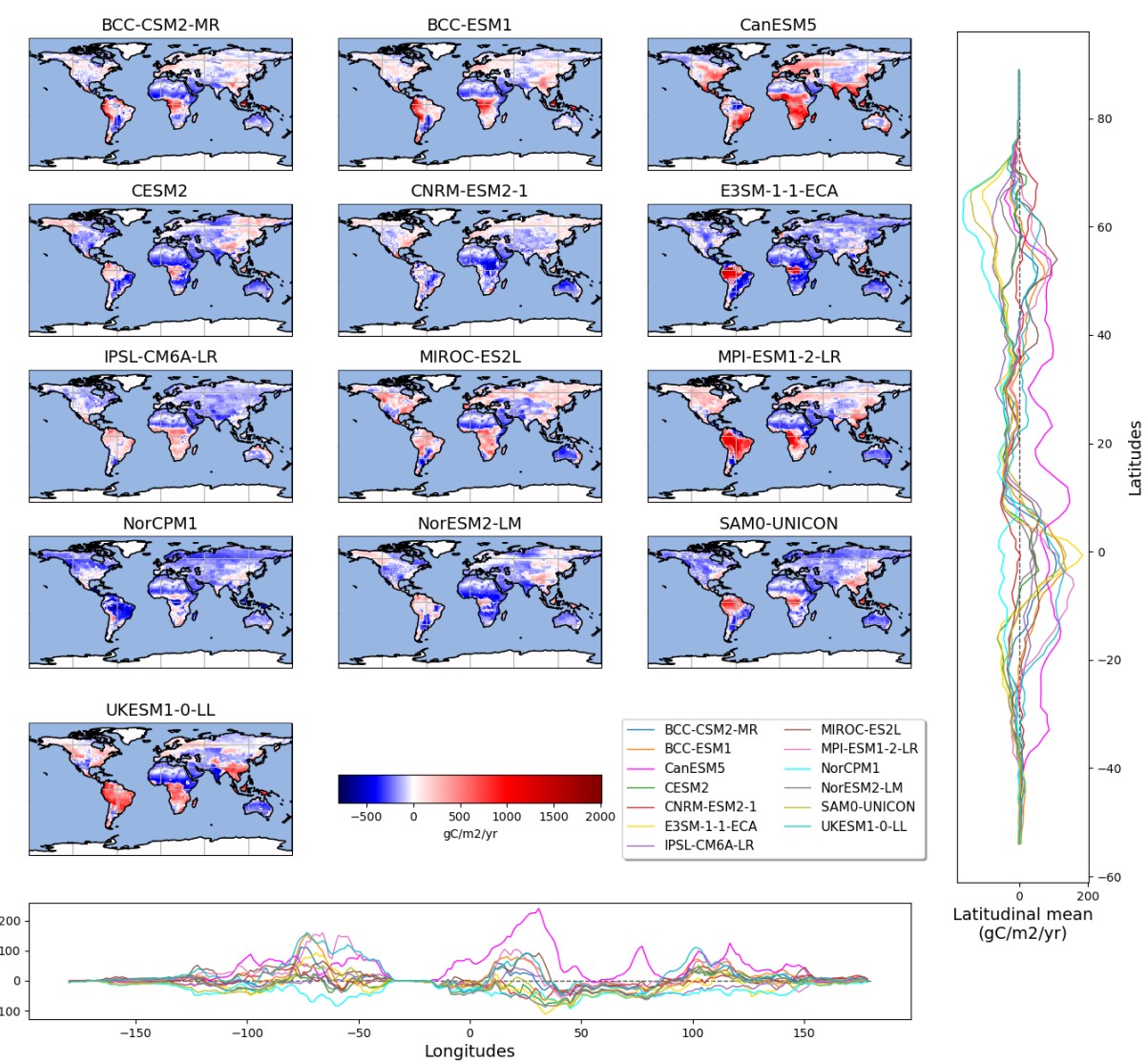

**Fig. 5. Spatially distributed residuals of CMIP6 ESMs predictions over the period 1990-2010 with respect to median observation products.**

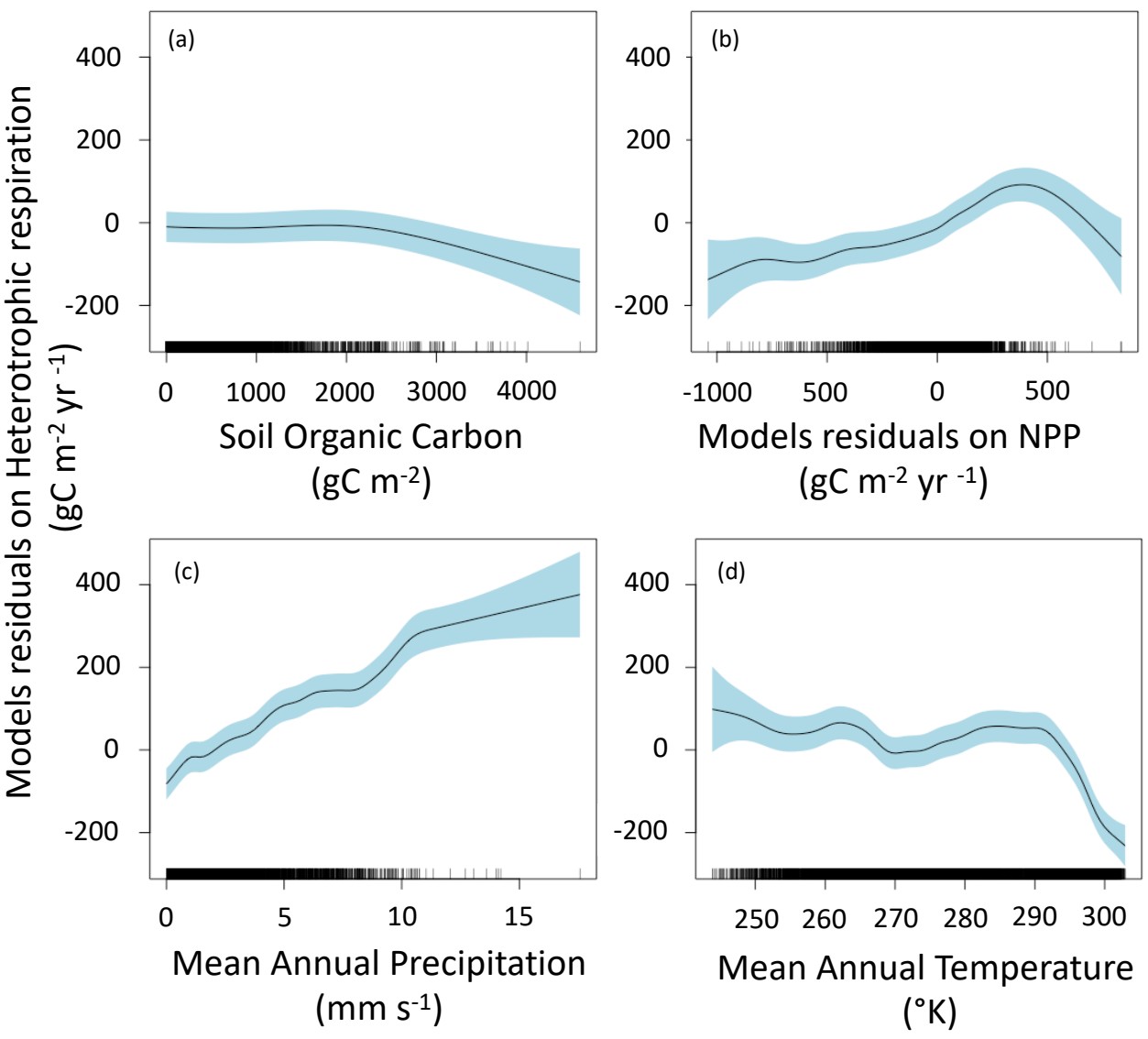

**Fig. 6. Median of ESMs residuals on soil heterotrophic respiration.** The residuals are explained by soil organic carbon (a), median of NPP residuals (b), mean annual precipitation (c) and mean annual temperature (d). Negative values mean model underestimation.

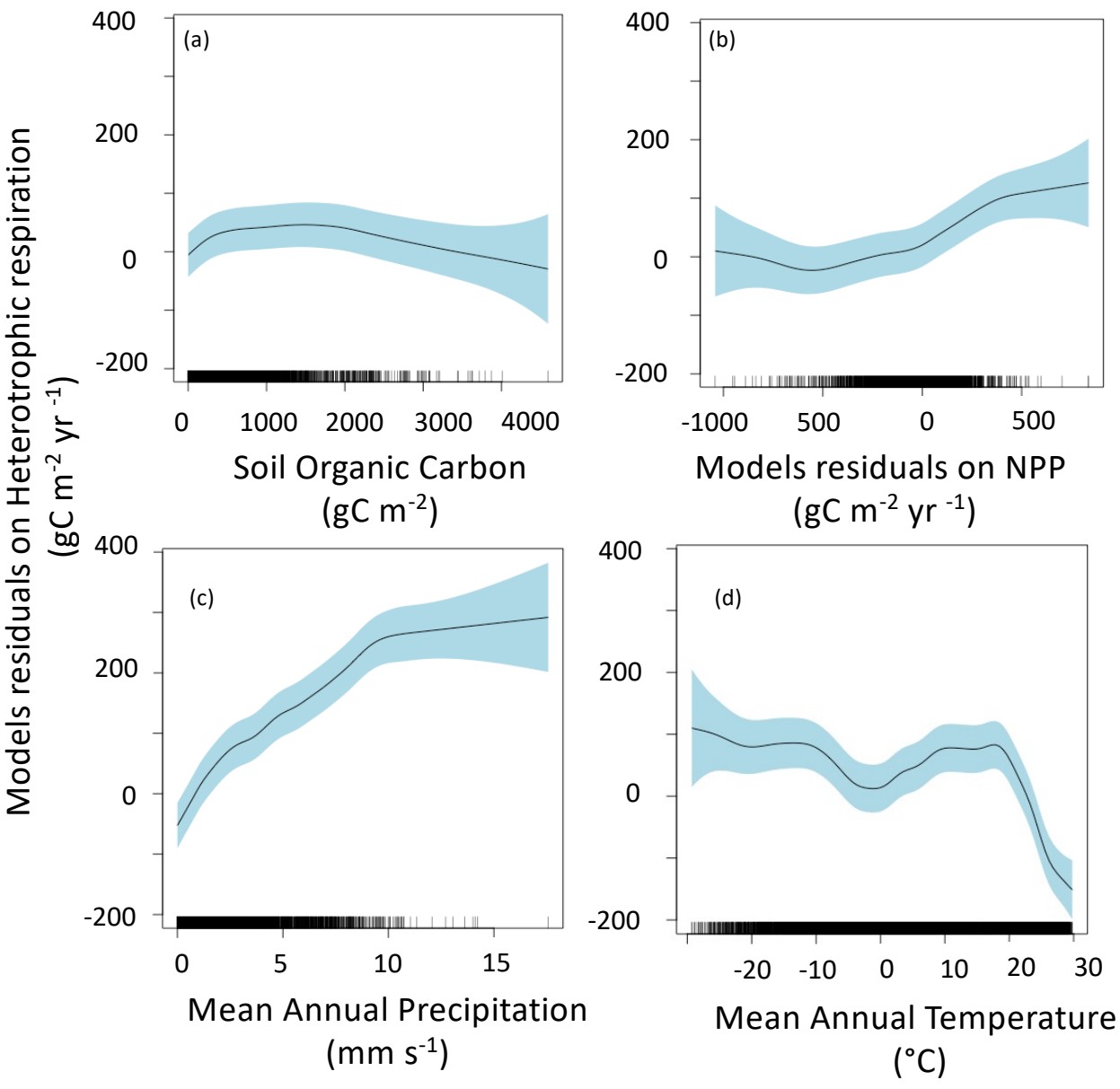

**Fig. 7. Mean of ESMs residuals on soil heterotrophic respiration.** The residuals are explained by soil organic carbon (a), mean of NPP residuals (b), mean annual precipitation (c) and mean annual temperature (d). Negative values mean model underestimation.

**Table 1. Evaluation metrics for the different Earth system models.**

| Models | BCC-CSM2-MR | BCC-ESM1 | CanESM5 | CESM2 | CNRM-ESM2-1 | E3SM-1-1-ECA | IPSL-CM6A-LR | MIROC-ES2L | MPI-ESM1-2-LR | NorCPM1 | NorESM2-LM | SAM0-UNICON | UKESM1-0-LL |
|---|---|---|---|---|---|---|---|---|---|---|---|---|---|
| % of grid cells within median ± mad | 43.9 | 44.2 | 30.4 | 32.6 | 42.0 | 29.9 | 36.2 | 25.0 | 30.2 | 27.1 | 36.0 | 25.1 | 27.7 |
| RMSE | 224.4 | 229.0 | 345.1 | 199.2 | 171.1 | 281.4 | 170.9 | 229.3 | 314.0 | 212.3 | 187.7 | 231.9 | 302.1 |
| R2 | 0.75 | 0.78 | 0.80 | 0.72 | 0.79 | 0.57 | 0.79 | 0.82 | 0.73 | 0.68 | 0.74 | 0.69 | 0.75 |

**Table 2. Summary of the main features proposed in this study to improve the heterotrophic respiration fluxes in Earth system models.**

| | Main features to improve in the next ESM generation |
|---|---|
| NPP residues | Improving plant inputs through NPP is key to improve heterotrophic respiration by implementing N cycle for instance. |
| MAT | Dynamic and/or spatialized temperature sensitivity parameters such as Q10. |
| MAP | Improving the soil moisture functions using bell shape functions for instance. |
| SOC | More constrained parameters such as CUE and/or residence times. |
