# Peer review of "Spatial biases reduce the ability of earth system models to simulate soil heterotrophic respiration fluxes"

_EGUsphere, 2023_

## Author Comment (AC2)

*Answer to comments from the reviewer #1.*

*We thank reviewer for the constructive evaluation of the manuscript. Please find below our answers to questions/comments. Comments from the reviewer were left intentionally in this document and written in roman font. Our answers are written in italics.*

This study provides useful analysis on Rh within CMIP6 ESMs against observational datasets and provides useful direction in future development of ESMs. It is a study that is well suited for publication in Biogesciences. Overall, the study is clear and well written, however some improved integration with existing studies and increased detail on the caveats will improve the study.

*Thank you for the positive comment. The revised version of the manuscript will be improved including your suggestions*

**Major comments**

Throughout the study it is referred to as being the 'first' to investigate heterotrophic respiration (Rh) in Earth system models (ESMs), though this is not the case. This study is still novel, however wording needs to be addressed here to include how this study fits in with the existing literature.

For example, in the abstract: "*capacity of Earth System Models (ESMs) to reproduce this flux has never been evaluated*" and "*for the first time*". Also, Line 182 in Discussion.

Relevant existing studies include:

1. Shao et al., 2013. This study evaluates Rh in CMIP5 ESMs against observational datasets (*Soil microbial respiration from observations and Earth System Models*).
2. Varney et al., 2022. This study focuses on soil carbon and has been cited, however spatial evaluation of soil carbon turnover (Cs / Rh) is included, and tables of global Rh values in CMIP6 and CMIP5 ESMs against observational dataset (Tables A1 and A2).

*Yes, this true we have been a bit clumsy in the wording, this has been corrected in several part of the revised version (see below for some examples).*

Line 183 – It is unclear to the reader what is meant here and there is no citation to back up this statement or to add clarity. Why is it that previously Rh in ESMs could only be constrained by NEE or ecosystem respiration? If the reason is lack of observational datasets, there are older soil respiration datasets (such as Raich et al., 2002 mentioned)? Plus, existing evaluation study on Rh in CMIP5 ESMs? Please expand on why this is the case or change the motivation behind the sentence.

*We clarified our point as following: "Indeed, previous dataset were not gridded and so far spatial pattern of heterotrophic respiration in ESMs could only by constraint indirectly by*

*constraining other C fluxes including heterotrophic respiration such as net ecosystem exchange fluxes or through ecosystem respiration in which heterotrophic respiration is just one component the other being the autotrophic respiration* (Stoy et al., 2013)*."*

Line 253 – Similar point here.

*We also rephrase to clarify : "Our study showed that despite previous ESMs evaluation on heterotrophic respiration* (Shao et al., 2013)*, a few current ESMs are fairly representing the total heterotrophic respiration flux but …"*

Line 36 – This sentence states that Rh has not been well incorporated into ESMs. If this is the case but this is the first study to evaluate this, how do we know? References need to be included here to back up this statement.

*We rephrase to clarify : "Despite the importance of heterotrophic respiration fluxes, the scheme representing this flux in ESMs, which aim to simulate the most important drivers of the earth's climate system, are currently challenged because important drivers are missing* (Huang et al., 2021; Wieder et al., 2015) *but the proposed new schemes lacks of sufficient evaluation on long term time series (Le Noë et al., 2023). Thus, how accurate are the prediction of ESMs for heterotrophic respiration fluxes is a key question to well constraint the carbon climate feedbacks in ESMs."*

Line 189 – The study notes large discrepancies in the observational datasets and is presented as an issue which needs to be addressed in the future. It would be beneficial to see more direct comparisons of the observational datasets. I think a useful addition to either an Appendix or Supplementary material would be comparing the observational datasets, potentially a correlation coefficient between them? I know maps of each are included in Fig. 3, but a quantification or difference map would be useful to see where there is more agreement or less agreement between them.

*In the revised version we have added difference maps between the products in the supplementary materials.*

It has previously been shown that the Hashimoto et al., 2015 dataset has an arbitrary maximum respiration level (see Supplementary Fig. 4 in Varney et al., 2020), which was shown in the same figure to not appear in additional respiration datasets. I think this point found here should be acknowledged and think about whether this could impact your residual results. Potentially the underestimation of Rh at high temperatures (Fig. 4)?

1. Varney, R.M., Chadburn, S.E., Friedlingstein, P. et al., A spatial emergent constraint on the sensitivity of soil carbon turnover to global warming. Nature Communications. **11**, 5544 (2020). https://doi.org/10.1038/s41467-020-19208-8.

*Very good point, indeed this is an interesting suggestion and we add information on the revised version: "… and then a sudden underestimation for warm temperatures above 290K corresponding to tropical and dry climate zones. This sudden underestimation might be*

*explained by the an arbitrary maximum respiration level observed in this dataset and identified as the result of the temperature-dependence of soil respiration used by* Hashimoto et al., (2015)  (Varney et al., 2020)*. Such bias can therefore should be a consequence of the observation-based products used here rather than a real ESMs bias."*

Line 139 states that the observational data and ESM data Rh means are close in Boreal regions. However, on line 163 it is stated that Rh is underestimated by ESMs for soils rich in carbon (which tend to be boreal regions). Any idea why this is the case?

*Indeed it might sounds surprising but two points may explain this. First, some peatlands are also in the tropic and because of the temperature conditions soil heterotrophic respiration may be higher and therefore impact more the results. Secondly, in boreal regions soils are carbon rich but temperature is cold and the bias explained by soil organic carbon can be compensated by the bias due to temperature that goes in the opposite direction (Fig. 4).*

Paragraph from Line 140 – only tropics and temperate regions mentioned, what about the northern latitudes?

*We added information in the revised version. "Models perform relatively well in temperate regions with for instance bias close to 0 gC m-2 yr-1 for BCC-ESM-1 over North America and Europe. Important discrepancies were observed for boreal regions with some models largely underestimating the heterotrophic respiration fluxes (e.g. NorCPM1 or SAM0-UNICON) and other overestimating the fluxes (MPI-ESM1-2-LR). The BCC models (BCC-CSM2-MR and BCC-ESM1) were performing quite well over this region.*

The use of the ESM and observational median is used throughout this study. I was wondering whether as it is not known which dataset or model is 'better', a mean value would give equal waiting to each, so could be a fairer metric. Does redoing the analysis with the mean instead affect the results? Especially spatially in regions where the datasets disagree more (Fig. 3)? If it does make a difference, it might be worth thinking about which is better for what you are trying to show or including in the Supplementary Material.

*We tried earlier to work with means and it did not change drastically the results. We decided to present medians instead of means because it was more adapted to small size populations. In the supplementary material of the revised version, we will present similar analysis with means instead of medians. You can find below for instance a comparison of the products means and medians. The mean and the median have similar patterns but the heterotrophic respiration is higher in the tropic with the mean because the weight of the Konnings et al products is higher when calculating the mean.*

**Median of the three spatially averaged observations datasets**

[Figure]

[Figure]

gC/m2/yr

**Mean of the three spatially averaged observations datasets**

[Figure]

gC/m2/yr

**Absolute difference mean - median of observations**

[Figure]

[Figure]

gC/m2/yr

**Relative difference (%)  mean - median of observations**

[Figure]

[Figure]

gC/m2/yr

Line 202 – The temperature sensitivity of soil carbon turnover time (Cs / Rh) has been previously investigated in similar ESMs, including discussion on variable Q10s spatially and a constraint on effective Q10 in ESMs (Koven et al., 2017 and Varney et al., 2020). This might link with some of the discussion in this paragraph.

1. Koven, C., Hugelius, G., Lawrence, D. et al., Higher climatological temperature sensitivity of soil carbon in cold than warm climates. Nature Climate Change. **7**, 817–822 (2017). https://doi.org/10.1038/nclimate3421.
2. Varney, R.M., Chadburn, S.E., Friedlingstein, P. et al,. A spatial emergent constraint on the sensitivity of soil carbon turnover to global warming. Nature Communications. **11**, 5544 (2020). https://doi.org/10.1038/s41467-020-19208-8.

*We modified the text in the revised version: "… with fixed parameters not dynamic and not spatially distributed (Ito et al., 2020).  Previous studies suggested that a spatially distributed Q10 constrained on observations would be an important step to improve ESMs (Koven et al., 2017; Varney et al., 2020). Our results are online with this statement and suggest that having more flexible Q10 parameters may help to improve ESMs capacities to reproduce observation-derived products of heterotrophic respiration fluxes.*

Line 203 – Could the underestimation in these regions be due to little or no soil carbon in these regions within ESMs (Varney et al. 2022)?

*Exact, we added this information : "Our study also showed that mean annual temperature is an important driver of the ESM residuals in particular for hot regions with large underestimations of the flux. It probably corresponds to very arid regions since for most of the ESMs, heterotrophic respiration fluxes from regions like Australia, Middle East or Northern Africa tend to be underestimated. Nevertheless, the underestimation observed in these regions can be also due to reduced C inputs and low SOC stocks reducing mechanically the heterotrophic respiration fluxes."*

Line 209 – I would also include a more recent reference, for example, Todd-Brown et al., 2018 (*Field-warmed soil carbon changes imply high 21st-century modeling uncertainty*). In this study Q10 values are derived and the sensitivity of ESMs to this parameter is investigated.

*The Todd-Brown et al. reference will be added in the revised version.*

**Minor Comments**

***All the minor comments will be considered in the revised version.***

Abstract – I would include that you are looking at CMIP6 ESMs here as I had to skim to the end of the introduction to check this, and it is useful to know upfront.

Line 31 / Line 188 – Update Friedlingstein et al., 2020 reference to Friedlingstein et al., 2022. As this is the most up to date Global Carbon Budget paper.

Line 66 – I don't think this sentence makes sense ", *which were used to derived two observation products we used.*" I think it should be "*dervive the*", rather than "*derived*".

Line 114 – The acronym AIC is used in this study, but it is not defined. I would at least add a sentence in the Methods to describe what this term measures.

Line 155 – Ito et al., 2020 is cited here, however the first order kinetics of decomposition is not discussed in this study that I can see. Todd-Brown et al. 2013 and Varney et al. 2022 include information and discussion about ESM decomposition dependencies to temperature and precipitation.

Line 160 – "*Since the drivers are ...*" might be worth changing to "*Since the **main** drivers are ...*" as many factors affecting respiration, as stated in your conclusions (Schmidt et al., 2011).

1. Schmidt, M., Torn, M., Abiven, S. et al., Persistence of soil organic matter as an ecosystem property. Nature. **478**, 49–56 (2011). https://doi.org/10.1038/nature10386.

Line 178 – Maybe better to present temperatures in degrees C rather than K in European journal, and better relates to how 1.5C / 2C targets are often presented.

Line 183 – This sentence might change due to an above comment, but there is a typo. "*by constrtaint*" should read "*be constrained*".

Line 210 – Do you mean Figure 4c here? I would include this in brackets so reader can be reminded where this result came from.

*References cited*

Hashimoto, S., Carvalhais, N., Ito, A., Migliavacca, M., Nishina, K., and Reichstein, M.: Global spatiotemporal distribution of soil respiration modeled using a global database, Biogeosciences, 12, 4121–4132, https://doi.org/10.5194/bg-12-4121-2015, 2015.

Huang, Y., Guenet, B., Wang, Y. L., and Ciais, P.: Global Simulation and Evaluation of Soil Organic Matter and Microbial Carbon and Nitrogen Stocks Using the Microbial Decomposition Model ORCHIMIC v2.0, Global Biogeochem. Cycles, 35, 1–20, https://doi.org/10.1029/2020GB006836, 2021.

Koven, C. D., Hugelius, G., Lawrence, D. M., and Wieder, W. R.: Higher climatological temperature sensitivity of soil carbon in cold than warm climates, Nat. Clim. Chang., 7, 817–822, https://doi.org/10.1038/nclimate3421, 2017.

Le Noë, J., Manzoni, S., Abramoff, R. Z., Bruni, E., Cardinael, R., Ciais, P., Chenu, C., Clivot, H., Derrien, D., Ferchaud, F., Garnier, P., Goll, D., Lashermes, G., Martin, M., Rasse, D. P., Rees, F., Sainte-Marie, J., Salmon, E., Schiedung, M., Schimel, J., Wieder, W. R., Abiven, S., Barré, P., Cécillon, L., and Guenet, B.: Soil organic carbon models need more independent time-series validation for reliable predictions, Commun. Earth Environ., 1–8, https://doi.org/10.1038/s43247-023-00830-5, 2023.

Shao, P., Zeng, X., Moore, D. J. P., and Zeng, X.: Soil microbial respiration from observations and Earth System Models, Environ. Res. Lett., 8, https://doi.org/10.1088/1748-9326/8/3/034034, 2013.

Stoy, P. C., Dietze, M. C., Richardson, A. D., Vargas, R., Barr, A. G., Anderson, R. S., Arain, M. A., Baker, I. T., Black, T. A., Chen, J. M., Cook, R. B., Gough, C. M., Grant, R. F., Hollinger, D. Y., Izaurralde, R. C., Kucharik, C. J., Lafleur, P., Law, B. E., Liu, S., Lokupitiya, E., Luo, Y., Munger, J. W., Peng, C., Poulter, B., Price, D. T., Ricciuto, D. M., Riley, W. J., Sahoo, A. K., Schaefer, K., Schwalm, C. R., Tian, H., Verbeeck, H., and Weng, E.: Evaluating the agreement between measurements and models of net ecosystem exchange at different times and timescales using wavelet coherence: An example using data from the North American Carbon Program Site-Level Interim Synthesis, Biogeosciences, 10, 6893–6909, https://doi.org/10.5194/bg-10-6893-2013, 2013.

Varney, R. M., Chadburn, S. E., Friedlingstein, P., Koven, C. D., Hugelius, G., Cox, P. M., and Burke, E. J.: soil carbon turnover to global warming, Nat. Commun., 4–11, https://doi.org/10.1038/s41467-020-19208-8, 2020.

Wieder, W. R., Allison, S. D., Davidson, E. a, Georgiou, K., Hararuk, O., He, Y., Hopkins, F., Luo, Y., Smith, M., Sulman, B. N., Todd-Brown, K. E. O., Wang, Y., Xia, J., and Xu, X.: Explicitly representing soil microbial processes in Earth system models, Global Biogeochem. Cycles, 29, https://doi.org/10.1002/2015GB005188, 2015.

---

## Author Comment (AC3)

*Answer to comments from the reviewer #2.*

*We thank reviewer for the constructive evaluation of the manuscript. Please find below our answers to questions/comments. Comments from the reviewer were left intentionally in this document and written in roman font. Our answers are written in italics.*

**Main comments**

This article presents an evaluation of global-scale heterotrophic respiration (Rh) from CMIP6 output in comparison to three different observation-based data products.

The main conclusion presented by the authors is that even though the global aggregated Rh agrees well between models and the data products, the models 'fail' at reproducing spatial patterns. The authors also provide a list of well-known mechanisms that influence soil respiration and advocate for their inclusion in new versions of the models.

Although in general I agree with the importance of model evaluation studies, I find little incremental value in this analysis. Despite the author's claim of priority, other studies have already made comparisons between ESM output and Rh data products, pointing out disagreements (see comments and references from other reviewers). The list of potential mechanisms to be included in a new generation of models, presented in the Discussion, are well-known mechanisms that influence soil carbon dynamics and Rh, and this discussion is relatively shallow regarding more relevant modeling topics such as the type of functions that should be implemented and how to obtain parameters for those new functions at the global scale. The analysis of residuals and their relation to other variables is helpful in providing some clues about the importance of these different processes, but without a more clear and systematic analysis of different mathematical functions to be implemented in ESMs, there are no elements for modeling teams to make decisions about what new functions to implement and how to obtain their parameters. For instance, this analysis identified a major discrepancy between residuals of Rh and precipitation, and the authors advocate the inclusion of hump-shaped functions in models, which is something that has been previously said (e.g., Moyano et al. 2013, Davidson et al. 2014). There are a number of such functions proposed in the literature (Sierra et al. 2015), and a more relevant discussion would be which of those functions are more relevant at the grid-size level of an ESM, and what type of observations should be used to obtain parameter values for these functions, or whether one single set of parameters should be used at the global scale or whether they should change spatially and temporally. Although I am not trying to convince the authors that they should add this discussion here, I feel that without a more in depth analysis, there is little new value in the present study.

*We thank the reviewer for the constructive comment. In the revised version we will carefully explain that indeed this is an incremental analysis but we still think our study is useful and worthy of publication for three main reasons:*

1.  *The analysis previously published was done on the CMIP5 models generation* (Shao et al., 2013) *and this paper focuses on the CMIP6 generation. Regarding the importance of the ESMs and their impact on others sciences, using results from the CMIP6 exercise is highly important to evaluate each model generations and share the results with the scientific community.*
2.  *Our study is novel because we take advantage of the gridded products that were not available before to better understand the spatial pattern of the heterotrophic respiration flux and how it is represented in the new ESMs generation.*
3.  *We used a model residue approach to disentangle the main effect and this was not used before. It helps to show that bias induced by the precipitation response is at least as important as those provide by temperature response.*

*Regarding the existing hump-shaped functions, it has been suggested before indeed but never done in ESMs and we consider that suggesting to the ESM developer community that some solutions might exist to solve the bias we identified is useful. Nevertheless, we agree that a more in-depth discussion might be useful.*

*In the revised version, we will add : "Implementing this bell-shaped function approach is necessary to accurately represent the soil organic carbon stock of peatland in some land surface schemes used by ESMs (Qiu et al., 2019). The approach proposed by Moyano et al. (2012) seems well adapted to ESMs constraint since the author proposed several versions of the bell-shaped function and did the effort to define one function using drivers that are included in ESMs (the model 2 in Moyano et al., (2012)). The model including bulk density might perform better but bulk density is not calculated by ESMs and consequently such approach is hardly implementable in ESMs. Other approaches have been proposed in the literature (Davidson et al., 2014; Sierra et al., 2014) but the solutions proposed are mostly based on Michaelis-Menten function whereas most of the ESMs used first order kinetics approach to describe SOM decomposition. Moreover, alternative solutions are based on $O_2$ diffusion which is more mechanistic but more difficult to implement in an ESM compared to a more empirical solution as proposed by Moyano et al. (2012). Gas diffusion implementation at the spatial resolution of ESMs is quite challenging because it depends on drivers highly variables at small scales."*

In addition, there are other topics of model evaluation that are very relevant for this study that are not discussed at all. One topic is the use of objective metrics to characterize distance between model output and data products. The authors claim that the models 'fail' to reproduce spatial patterns, but a definition of 'failure' is not provided, nor a measure of distance or probability of model output to lay in some rejection zone. A more formal analysis would be required to assess how far the model output is with respect to data-products, which are also uncertain. Throughout the manuscript the authors use the three data products as error free, but it is well-known that these products are also subjected to biases and errors. Despite their growing size, Rh databases still lack comprehensive coverage in some key regions such as the tropics. If all the models would agree well with a biased data-product, we would be very misled in our carbon-climate projections!

*We obviously agree that all the observation-based products are somehow uncertain and this is why we decided to use several of them in this study. We do not have all the information to*

*calculate an in-depth error propagation but in the revised version, we will better quantify the uncertainties by:*

1. *Calculating the median absolute deviation (MAD) of the median of the three products*
2. *Ranking the ESMs by the number of pixels that are within the MAD*
3. *Calculate RMSE for each ESM.*

Another topic of relevance is the issue of spatial aggregation in soil respiration estimates. Since the 1990s, there has been a discussion on how to deal with aggregation errors in estimates of Rh at ecosystem and global scales (Kicklighter et al. 1994, Rastteter et al. 1992). The authors downscaled the CMIP6 output to a common spatial resolution, but it is not clear how this 'dis-aggregation' would affect uncertainties and biases.

*The method we used is conservative and the main problem we faced during the regriding step was to take into account that the land fraction changed because of the regriding. The difficulties were similar for model outputs and for observation-based products. Once we corrected for the land fraction change the regriding effect was quite limited. For instance, our regrided products estimate the total heterotrophic respiration to be 50, 43 and 51 PgC yr-1 for Warner et al, Konings et al and Hashimoto et al., products respectively whereas in the original publications they authors estimated 49.7, 43.6 and 51 Pg PgC yr-1 for Warner et al, Konings et al and Hashimoto et al., products respectively. The spatial distribution was also not hardly affected by the regriding (see the figure below using the Konings et al. product to illustrate).*

[Figure]

Mean Rh spatial distribution over 2010-2012 from the Konings et al., (2019) product –original (46x72, top panel) vs regrided (128x256, bottom panel).

In summary, although the results presented here are interesting to explore differences between CMIP6 Rh output with respect to observation-based data products, the authors make claims about scientific priority/novelty and 'failure' of the models that are poorly supported.

**Minor comments**

- L37-40. What do you mean by that these fluxes are not well characterized? Do you mean 'evaluated' instead of 'characterized'? What has been done with plant and ocean fluxes that has not been done with Rh?

*We rephrase to clarify : "Despite the importance of heterotrophic respiration fluxes, the scheme representing this flux in ESMs, which aim to simulate the most important drivers of the earth's climate system, are currently challenged because important drivers are missing (Huang et al., 2021; Wieder et al., 2015) but the proposed new schemes lacks of sufficient evaluation on long term time series (Le Noë et al., 2023). Thus, how accurate are the prediction of ESMs for heterotrophic respiration fluxes is a key question to well constraint the carbon climate feedbacks in ESMs."*

- L94. Please provide more details about 'cdo remapdis (nco module)'. What is this? A software, a package of a programing language? Can you provide a reference?

*We added this information: "The Climate Data Operators (CDO) software is a collection of multiple operators for standard processing of climate and forecast model data. The operators include simple functions (statistical and arithmetic) to be used for data selection, subsampling, and spatial interpolation."*

- Section 2.5. This paragraph is very difficult to understand. I get the general idea of the analysis, but I can't understand well the specific details. Please consider rewriting this section, adding more details for each step, adding some equations about how the medians and model differences were obtained, and maybe a figure describing the different steps.

*We rewrote this section: "We defined here the ESM's model residuals as median of the difference between each single CMIP6's model output and the observation-based products median calculated for each grid cell. The ESM's model residuals were calculated in three steps: (i) we calculated first the median for each cell using the three observation-derived products. We consider this median as our best-estimate. (ii) Then, we calculated the difference between each CMIP6's model output and our best-estimate for each grid cell. (iii) Finally, we calculated the ESM's model residuals as the median of this difference.*

*Using the ESM's model residuals, we performed a statistical analysis to identify the main drivers. We proceed with a two-step methodology. First, we compared several linear generalized least square models with different spatial structures (gaussian, exponential, spherical, linear or rational (gls package, (Venables and Ripley, 2002))) and without spatial structures to estimate the effect of spatial correlation. Based on AIC values we selected the rational quadratic spatial correlation structure that had the smallest AIC values for the second step of the analysis. Then, we used generalized additive mixed model with ESM's model residuals as variable to explain and mean annual temperature (MAT), mean annual precipitation (MAP), observation derived SOC, ESM's model residuals on NPP and lithology as predictors variables. MAT and MAP are derived from the Global Soil Wetness Project Phase 3 (GSWP3) reanalysis (http://hydro.iis.u-tokyo.ac.jp/GSWP3/ last access: April 5 2022). SOC was taken from the Soilgrid250m product(Hengl et al., 2017).  ESM's model residuals on NPP*

*are calculated as the median of the difference between ESM's NPP and NPP from the global inventory monitoring and modelling studies group (GIMMS). Lithology maps from the global lithological map (GLiM) (Hartmann and Moosdorf, 2012) was used but since lithology was not significant (p>0.05) and the model has a lower AIC without it was not included in the final generalized additive mixed model presented here. All statistical analysis were made using R v3.5 (R Core Team, 2018)."*

- L114. From what programing language is the gls package? Add a reference.

*The gls package is from R as explained at the end of the paragraph. We added the Venables, W.N. and Ripley, B.D. (2002) "Modern Applied Statistics with S", 4th Edition, Springer-Verlag. Reference which is cited in the documentation of the function.*

- L140-141. The median of the mean across products? or the median of the residuals after fitting a statistical model? Legend of Fig 3 says that each map is a residual. Be more specific.

*We first calculated the median for each observation-based product for each grid cells. We this obtained our best-estimate spatially distributed. Then we calculated the residual for each model at each grid cell. We modified the text to clarify:*

*"To generate our best-estimate of heterotrophic respiration fluxes from the three observation-derived products we calculated the median for each cell. Thus, we obtained the spatially distributed best-estimate. At each grid cell, we then compared each ESM with the observation-derived products median (Fig. 3)."*

- L142. I'm not sure if 'overestimate' is the right word to use here. The comparison is not directly with measured data, but with the output of a model that was informed by data. The data-products may also include biases.

*We rephrase to clarify: "Compared to observation-based products, ESMs tend to overestimate heterotrophic respiration flux in tropical regions…"*

- L158-159. I still don't understand how the use of first-order rates in models is connected to the need to use the median of the residuals in this comparison. Can you explain this better?

*We modified in the revised version to clarify: "In order to improve predictions of heterotrophic respiration fluxes in future ESMs we need to understand the spatial biases we observed and determine their causes. To explore these biases, we performed a statistical analysis based on a generalized additive mixed model of the ESMs residuals defined as the median of the difference between each CMIP6's model output and the median of the observation-based products calculated in each grid cell (see online methods). ESMs share a very common approach based on first order kinetics with soil organic decomposition driven by soil moisture and temperature (Ito et al., 2020). This approach is derived from the very first attempts to describe soil organic decomposition with mathematical equations (Henin and Dupuis, 1945) and is still the most used to describe this process (Manzoni and Porporato, 2009; Wutzler et al., 2008). Since SOM decomposition schemes in ESMs are very similar, comparing each model individually can be redundant and not very informative and less generalizable. To allow broader conclusions and suggestions to improve ESMs performances,*

*we decided to perform the residual analysis on the ESMs median rather on each individual model."*

- L160-161. This set of drivers of Rh is well-know, even before Swift et al. (1979). I'm not sure why this single recent reference is relevant here.

*We used the Doetterl et al. (2015) study to support this claim because it was done at global scale with a very large dataset.*

- L160-163. The entire sentence is difficult to understand. Consider rewriting.

*We modified the sentence in the revised version: "The main drivers of heterotrophic respiration are soil carbon availability, soil moisture and temperature, carbon inputs and mineralogy (Doetterl et al., 2015). To explain our model residues we used soil organic carbon, net primary production residuals calculated using similar methods to heterotrophic respiration flux residuals, mean annual precipitation, mean annual temperature and lithology.*

**References**

E. A. Davidson, K. E. Savage, and A. C. Finzi. A big- microsite framework for soil carbon modeling. Global Change Biology, 20(12):3610–3620, 2014.

D. W. Kicklighter, J. M. Melillo, W. T. Peterjohn, E. B. Rastetter, A. D. McGuire, P. A. Steudler, and J. D. Aber. Aspects of spatial and temporal aggregation in estimating regional carbon dioxide fluxes from temperate forest soils. J. Geophys. Res., 99(D1):1303–1315, 1994.

F. E. Moyano, S. Manzoni, and C. Chenu. Responses of soil heterotrophic respiration to moisture availability: An exploration of processes and models. Soil Biology and Biochemistry, 59(0):72 – 85, 2013.

Rastetter, King, Cosby, Hornberger, O'Neill, and Hobbie] E. B. Rastetter, A. W. King, B. J. Cosby, G. M. Hornberger, R. V. O'Neill, and J. E. Hobbie. Aggregating fine-scale ecological knowledge to model coarser-scale attributes of ecosystems. Ecological Applications, 2(1):55–70, 1992.

C. A. Sierra, S. E. Trumbore, E. A. Davidson, S. Vicca, and I. Janssens. Sensitivity of decomposition rates of soil organic matter with respect to simultaneous changes in temperature and moisture. Journal of Advances in Modeling Earth Systems, 7(1):335–356, 2015.

M. J. Swift, O. W. Heal, and J. M. Anderson. Decomposition in terrestrial ecosystems. University of California Press, Berkeley, 1979.

---

## Author Response (AR2)

In this new version, the authors made important clarifications that have improved the article considerably. Nevertheless, there are a few other issues in this new version that require attention:

- The abstract and other parts of the manuscript still make major claims of priority. I understand that other studies in the past didn't work with CMIP6 and gridded observational products, but it still feels disrespectful regarding the work that other colleagues have done in the past. The abstract does not have the level of detail for a casual reader to understand what is really novel here and why this claim of originality.

*The abstract has been rewritten with more details added*

- The description of the data analysis in three different steps (section 2.5) is helpful, but it still lacks detail. For example, step 3 is still ambiguous regarding on how those medians were calculated. By pooling all the residuals of all models or of each individual model? The most appropriate way, which is followed by most authors in other analyses, is to present equations with the respective indices for gridcells and models. I recommend the authors to add the specific equations here, otherwise their description will still suffer from ambiguities.

*We added equations detailing the calculation procedure.*

- The new text on the moisture dependent functions is helpful, but there is still a problem in the interpretation of the model of Davidson et al. (2014). The authors seem to imply that this model cannot be implemented in ESMs because most of them are based on first order kinetics while Davidson's is based on Michaelis-Menten kinetics. This is a misunderstanding. Davidson's model is a function for a rate modifier, is not a system of differential equations with interacting substrates that would lead to nonlinear kinetics. The rate modifier function is expressed as what you would called Michaelis-Menten kinetics, but others would simply call a logistic function. It is only used to obtain a rate modifier, and it can be incorporated in ESMs as any other rate modifier. It is true that that oxygen and soil moisture dynamics must be modeled explicitly in the ESM, but that's different problem.

*This part was removed and we only mention the oxygen limitation in the new version.*

- The conclusion section makes the case for the improvement of models, but there's no word on improving the observation-based data products. Are we still assuming that they are perfect and we only need to improve the models? Your analysis showed that at least one data product may be biased at high temperatures. I think we still need to improve these products considerably.

*We fully agree with this point but since we are not observation derived products producers it is more difficult for us to provide useful point on this aspect. Nevertheless we added some information in the conclusion.*

- Table 1 is a welcome addition to the article. Can you add an extra row with the proportion of pixels within the median. The absolute number of pixels are difficult to grasp without an idea of the total number of pixels.

*Table 1 was modified with the proportion instead of the total number of grid cells.*

The new version, and in particular the new added text has a number of typos and small errors. Please read your manuscript more carefully before final submission. Some of the issues are:
- Ln 56, 'propose a way of improvement'?
- Ln 153, comma is missing? Consider rewriting.
- Ln 217, residues?
- Ln 271. 'constraint ... by constraining? Revise.
- Ln 313. 'mechanically'?

*Our co-author Phil Martin who's a Native speaker checked the English of the last version of the manuscript.*